# Positioning Theory in Education

**Sonia Martins Felix \***  **and Sikunder Ali**

Department of Teacher Education, Norwegian University of Science and Technology, 7491 Trondheim, Norway;
sikunder.ali@ntnu.no
* Correspondence: sonia.f.martins@ntnu.no

**Definition:** Positioning theory is a social theorization that aims to capture the dynamic analysis of conversations and discourses taking place in a social setting. Conversations as part of language assume interlocutors. As one engages in the interactive speech acts in the social setting, there comes the importance of interlocutors involved in these speech acts in creating a social reality, as language forms the knowledge of reality. Certain types of rights and duties can be observed in interactions between speakers and hearers in a social communicative context of interlocutors. The cluster of rights and duties, recognized in a certain social setting, can be termed as a position. One of the critical aspects is that positions are not always intentional or even conscious. Therefore, positioning theory has been redefined as a method of analysis with a focus on storylines. Storylines reveal implicit ascriptions and resistances of rights and duties through the performance of a variety of actions in a social setting where appropriateness of social acts are established and recognized by the participants engaged within the social situation. The education setting presents a dynamic situation where a variety of moral orders come into actions that set possibilities for different actors to engage in shifting positioning to accomplish certain educational actions. This entry presents the use of positioning theory in an educational setting.

**Keywords:** positioning theory; education; positions; moral orders; rights and duties; storylines





## 1. Introduction

Positioning theory studies the dynamics of positionings with interactive phenomena where different actors come under conversations by offering their storylines [1–3]. These storylines are then shaped by discursive acts that take place within specific social practice(s). In this connection, the contextual location of positioning theory can be traced back to the communicative nature of human actions. For example, Gregory Bateson (1965; 2000) [4,5] considers human actions as happening at different levels: individual, social and ecological. These actions could be influenced by cultural, religious and specific social practices that enable these human actions and meanings to take place. He emphasized the role of symmetry and complementarity that come into action under the mutual influence of human interactions by illustrating *Naven* play under local customs in New Guinea, where the roles of men and women were reversed in the rituals (as Naven) (see Bateson, 1965) [4]. He calls the interaction among individuals social and ecological as part of cybernetics, which goes beyond any of these three elements. The interactions among these three elements lead to the possibility of considering human actions as communicative actions. Watzlawick and Jackson (2010) [6] consider human communication as an integral part of human sense making. They [6] give the example of Descartes' *Cogito, ergo sum* (I think therefore I am) formula, which cannot make sense until there is a social context that creates a need to think. Thinking as an activity requires an action on the part of the other person to understand that as such. That is, human communicative actions are essentially social actions. This has been well illustrated by Jürgen Habermas (1981; 1990) [7,8] through his thesis on the constitution of reality through communicative actions for achieving cooperation with an emancipatory intent. Of course, these communicative actions and the constitution of reality

encompass both symmetric and asymmetric communications with attention to languages, social norms, customs and cultures that condition such communications. The asymmetry of communications within a social situation brings attention to the role of discursive norms and discourses that shape such actions and influence the power therein. The circulation of power and shifting points of power during communicative actions are considered to be critical in understanding the positions that different actors take within a social practice and, in turn, how linguistic structures constitute the social world.

Potter et al. (2015) [9] identifies the importance of understanding social practice in situ in order to see how different discourses in fact create and constrain possibilities for actors to participate within discourse in communicative situations. Here, Potter and his colleagues consider the importance of assuming discourse from the constructions perspective. They suggest that social actions are embedded in multiple discourses and these discourses have three characteristics: (1) functions (goal-oriented actions); (2) construction (that actions are constructed); and (3) variations (such constructed oriented actions make such variations possible). Here, discourses instead should be looked at as interpretive repertoires orienting towards achieving social actions within a possible range of variations that the discourses make available.

Moreover, social psychology and sociology also bring insight into social actions. Here, language, socializations, norms and values embedded within cultural practices orient social actions in a particular way. For example, Butler (1988) [10] analyzes how feminine women become through the repetitive actions and norms that format social realities, which then establishes gender roles and then re-enforces these roles through different norms and cultural practices as developed historically, politically and culturally sanctioned. Further, she points out the importance of challenging the established cultural categories as gender. Moreover, she raises the importance of understanding the effects that speech acts can bring on the particular actions of actors. John Austin (1962) [11] performed a systematic study of speech acts in social linguistics and identified different kinds of speech acts: (1) representative, which specifies acts such as assertions, statements, claims, suggestions and hypothesis; (2) commissive, which encompasses promises, oaths and pledges; (3) directive, which includes orders, requests, invitations and challenges; (4) declarations such as baptisms, arrests and legal actions as sentencing; (5) expressive, such as greetings, congratulations, apologies, etc.; and (6) verdictive, which includes rankings, appraisings and condoning. John Searle (1995) [12] extended this work and tried to articulate how social objects such as money become constituted through various social actions under collective intentions. In addition, collective intentions could be contrasted by individual intentions, and collective intentions cannot be reduced to individual intentions. However, individual intensions can be influenced by collective intentions.

The above research contexts with a focus on human actions, human communicative action, speech acts and the constitution of social actions through discourse and discursive acts that underpin norms, values and cultures responsible for these social actions have provided a ground on which the idea of positioning theory emerged in the 1980s. In the 1980s, the dominant influence of positivist psychology (based on survey methods or psychometric tools modeled on experimental psychology) was challenged by paying attention to the dynamic constitutions of social actions by foregrounding social practice in situ and actors' active involvement in shaping the social reality in which they found themselves.

Within this context, this entry seeks to give an overview of the positioning theory and the implications it can have on education, while using some examples.

Assuming language as a social practice refers to a mode of action (Austin, 1962; Fairclough, 1995; 2003) [11,13,14]. It means that social practice is always situated in the social mode of action happening in dialectical relationships where multiple actors come together to accomplish certain actions in interactive social conditions. Thus, considering any conversation with some interlocutors, this simple action implies that, through a 'speech-act,' we use a certain 'storyline' to position ourselves in the performative act of positioning.

Conversational episodes are fundamental units that shape social reality and that structure social interaction. The relation between positions, speech and other acts and storylines are captured through the positioning triangle as initially proposed by Van Langenhove and Harré (1999) [1], later adapted by McVee et al., 2011 [15] and by us recently (see Figure 1).

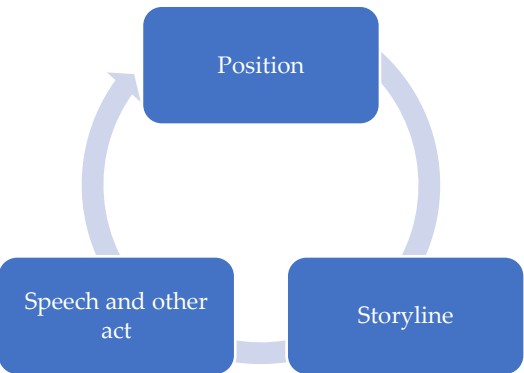

**Figure 1.** Aspects of social acts through positioning theory (inspired by the positioning triangle from McVee et al., 2011 [15]).

These three aspects (positions, speech acts and storylines) are fundamental in characterizing the complexity of social acts through positioning theory. We understand positions as something that is highly dynamic and that requires to take into consideration speech and other acts and storylines simultaneously. A more detailed explanation is provided below for each concept.

### 1.1. Positions

Positions can be defined as clusters of norms tacitly agreed by the actors in the context of interactions (Moghadam & Harré, 2008) [16]. In these interactions, rights and duties are being dynamically shared and shaped by the actors involved in certain conversational acts. Because each speech act belongs to a person, positions can be identified as ways to capture the dynamics aspects of selfhood (Kayi-Aydar, 2018) [17]. Relying on the idea of Tirado and Gálvez (2008) [18] about positioning, interaction is a discursive and narrative process which is situationally developed. In the context of education, teachers have a central role in educational acts. In order to understand the dynamic actions of teachers, it requires one to pay attention in terms of analyzing the discursive acts that they perform and are simultaneously conditioned by. These analyses must be conducted based on the active role that teachers take, especially within the discourse happening in the interactive educational setting where many actors are involved. The term position has been commonly used in social psychology and sociology (Hirvonen, 2016) [19].

### 1.2. Speech and Other Acts

This refers to the way people use the words to locate themselves or "words we say or write and any other meaningful communicative act" (McVee et al., 2011, p. 6) [15]. The speech act has been defined as performances when people interact [1] (Harré and Van Langenhove, 1999). Here, words bring effects and effects can be detected through the performance of actions that actors achieve through words. This dynamic interactive interchange between words and acts is demonstrated well through the works of John L Austen's *How to do things with words* (1962) [12] and of John Searle (1969) [20]. John Austin (1962) [11] gives an example of effects of words such as: I marry you as husband and wife. The effect of this speech act accomplishes a legal contract of marriage as legitimate and changes the situations of two individuals in transforming them as a married couple within a socio-legal power discourse in a society under the nexus of rights and duties.

*1.3. Storylines*

Storylines usually refer to "dynamic episodes or patterns that are created through speech act and positions" (McVee et al., 2011, p. 6) [15]. These patterns were previously defined as a "loose cluster of narrative conventions" (Harré & Moghaddam, 2003) [16]. Van Langenhove (2019) [21] refers to the fact that storylines are more likely to be introduced by some people. This indicates complex configurations (symmetric or asymmetric) and interplay between different storylines in each social setting/context where the social field of rights and duties are dynamically established, negotiated and resisted. For example, positioning theory has been found useful in highlighting people's rights and duties, showing how "failure" can be traced back to (i) individuals in classrooms (Anderson, 2009) [22]; or (ii) a group of people (teachers) showing/bestowing trust on each other (Felix & Ali, 2022) [23], as revealed through different storylines.

## 2. Why Positioning Theory?

Harré (2015) [24], in his presentation in a symposium in Bruges (8 July 2015), explains what the virtue of the existence of positioning theory is and considers three conceptions of social (interpersonal) essential actions:

1.  What you can do.
2.  What you do.
3.  What you are permitted or forbidden to do.

In an education setting/context, the perennial distinction between "what you can do" and "what you actually do" points towards anticipation: "What you are permitted or forbidden to do". This third domain is what Harré (2015) [24] called the province of positioning theory, and that was not captured in social psychology and sociology modeled on experimental positivist orientation where social reality was considered as static to be discovered, not seen as constructed. In education, positioning theory has been perceived as a tool to capture the complexity of educational action (Green et al., 2020) [25]. For example, in the case of the teachers, positioning theory allows one to see how they position themselves as indicated through speech acts, when they interact in social acts [1] (Harré & Van Langenhove, 1999) as educational actions. One important aspect is that, through positions, teachers are indicating ways in which they pursue their rights and duties in relation to certain aspects of social reality. That implies that, by paying attention to teachers' diverse positional acts in the educational setting they are in, one identifies the speech/acts that they are permitted to say or do. In other words, positioning theory allows one to seek what a person 'may do and may not do'. In this way, 'rights' and 'duties can point towards clusters of morals (normative acts)'. This means that rights and duties are immersed in clusters of morals, which helps us to look at what people believe or are told or are 'following into' and to which they are momentarily tied in what they say and do (Harré et al., 2009) [26]. Clusters of morals (normative acts) can also be culturally and locally determined. For example, broader cultural moral norms such as occidental or oriental could also shape local actions of actors. This leads one to consider the rights and obligations that are enacted in situ where teachers find themselves and how teachers position themselves in reacting and transforming their actions within these rights and duties, or the social field of actions. Similarly, such configurations of rights and duties can also be observed among the actors in situations such as economic activity of exchange and in social situations such as gender roles, responsibilities and moral actions that form conditions for realizing these actions.

Harré (2012) [3] defined an act as a social meaning of a certain action and speech as meaningful and intended performance. Positioning theory considers individuals. However, positions, as stated above, are comprised between individuals' storylines and speech acts and thus constitute the positioning triangle [1] (Van Langenhove & Harré, 1999). Identifying teachers' positions through positioning theory is a way to study local moral orders constrained by ways of exercising rights and duties through ways of speaking and acting [1] (Harré & van Langenhove, 1999). Moral orders can be characterized as

always shifting patterns of mutual and contestable rights and obligations through the way one speaks or acts [1] (Harré & Van Langenhove, 1999). In other words, moral orders are created between structures and persons through declarative speech acts with deontic powers where deontic refers to the deontic theory of ethics, which pays attention to duties and the actions that ought to be done. For example, any teaching site, such as the classroom, is located within socio-legal and cultural frames where many actors (students, teachers, headteacher, school administrator, parents, policy makers, etc.) come together in a certain normative contract (within multiplicity of moral orders) in which they perform actions, take dynamic positions and are engaged in a continual process of creating an educational act. Here, positioning theory is giving centrality in conversations and contributes to a non-deterministic aspect of the reality. Moreover, it helps to establish moral orders in a dynamic stability between different people's positions, i.e., the social force of what people say and do, through storylines [1] (Van Langenhove & Harré, 1999). In education, one can consider nested orders since it could constitute as a normative field in which people have to act and respond (Van Langenhove, 2017) [27].

### 3. Varieties of Moral Orders and Positioning Theory

Moral theories have been describing ways of understanding how people move through different stages in a fixed order (as inspired through Piaget's developmental stages), considering cognitive development (Kohlberg, 1981) [28]. Kohlberg (1981) [28] proposes six moral developmental stages: (1) obligation and punishment orientation; (2) individualism and exchange; (3) good interpersonal relationships; (4) maintaining a social order; (5) social contracts and individual rights; and (6) universal principles (actions geared towards social justice principles). These stages allow us to see how moral orientations guide/shape human actions when one faces moral dilemmas. Here, we consider that morality can be seen as a key towards the understanding of complexity of human behaviors or actions under constraints of varieties of moral orders (Van Langenhove, 2017) [27]. In a given time, people are located in a certain moral order as a consequence of the ever-shifting patterns of mutual rights and duties through discursive acts [1] (Harré & Van Langenhove, 1999). That is, moral orders could create possible fields of actions within interactions between rights and duties within a specific situation and social practice. This characterization of a variety of moral orders assumes the dynamic nature of moral values which come into action in specific situations. This dynamic nature of the variety of moral order does not assume the sequential nature of moral values, as articulated by Kohlberg [28]. However, moral values in terms of stages could be a theoretical resource for analyzing different positionings that actors take in invoking their positions in a particular social setting through their storylines.

Relying on this idea, Van Langenhove (2017) [27] presents a variety of moral orders, based on the idea that different structures operate as moral orders. Harré (1984) [28] was the first who proposed a systematic theory of moral order as an effort to describe rights and duties of individuals through positioning theory. Harré [28] has been arguing for developing positioning theory as an alternative to the concept of *role* in the field of social psychology (McVee et al., 2019) [29]. One important aspect in education is that it serves a "variety of moral orders" (Van Langenhove, 2017) [27] and, thus, can be characterized in a variety of moral orders with the awareness that people's rights and duties differ from situations and context as synthesized in Table 1.

**Table 1.** Summary of variety of moral orders as articulated by Van Langenhove, 2017 [27].

| Moral Orders Defined by Van Langenhoven (2017) [27] | Characterization |
|---|---|
| Cultural moral order | Very general nature, i.e., cultural located in a certain civilization.<br>Includes moral opinions that are part of religious or secular codes. |
| Legal moral order | Territory of specific states or regions are bounded (in a strong juridical sense). It can be delimited by international order (through transnational organizations as UN, EU (in a more consensual framework such as Sustainable Development Goals, human rights issues, protocols, mechanisms for reaching agreements and resolving disagreements and conflicts under certain moral orders). Emphasis on telling you what to do/what not to do. |
| Institutional moral order | When individuals take up memberships in organizations (whole set of rules). It covers a wide range of social things. |
| Conversational moral order | Conversations occurring between people who are engaged. This means that local moral orders are in play that are certainly part of general moral order. |
| Personal moral order | When individuals have conversations with themselves. Presupposes internal dialogues where one deliberates about what is right or wrong to do. For example, when using the indexical word "I," individuals are creating their moral individuality in confrontation with others (Harré & Gillett, 1994) [30]. |

Paying attention to the cultural, legal, conversational and personal dimensions of moral orders within education settings will allow us to see how complex educational action is taking place and how different actors are engaging in the struggle to define themselves and are seeking the legitimacy of their actions. Moreover, moral orders are responsible for setting up boundaries where changes in educational actions are framed at the local level by invoking archetypical values such as freedom, democracy, relevancy, connection of education with job market, progress of society, etc. (Popkewitz, 2008) [31]. Given that educational action is a mandated action agreed through the political action taken through the constitutional provision of education, this underpinning of legal conditions of education form the boundaries of actions that become possible within the school and classroom situation, and types of positions become available for different actors (teachers, students) to take to develop their capacities through the educational act. Here, positioning theory allows us to see how actors strategically relate to each other to achieve certain actions as achievable and certain actions as classified as irrelevant. For example, the democratic discourse of inclusion and diversity allows students with autism to be included/excluded within the regular classrooms so that they can be positioned as equals in educational setting of the classrooms/schools. Extra education resources, such as the availability of special educational teachers, are made available to provide the necessary potential support so that these students with autism could participate equally in the normal classrooms. Now, a cluster of critical questions can arise: How do autism students feel about their participation in a regular classroom? Do they really feel included in this regular classroom? Are they really positioned equally in the dynamic interactions in the classroom, especially when values are being placed on the idea of students to demonstrate their competency through their argumentative, reasoning linguistic communicative competence in a classroom situation? New reforms in education valorize more communicative competences and promote discourse of effective problem solvers through the expectation of students as competent communicators who actively solve the complex problems of life under changing conditions of society where multiple risks and uncertain conditions are norms. Considerations of

paying attention to dynamic conversations and personal actions that actors take within a social reality as classroom/school would allow us to trace moral orders that limit and open or restrict possibilities for actions in the educational setting. Moreover, positioning theory can allow us to trace positive and negative effects that power (as is expressed through conversations among actors entangled within variety of moral orders) or regulate the actions of people within educational settings. Here, power could be understood, not as a fixed entity that someone has due to the position that she or he possesses, but as how power is generated and regenerated within conversational settings under the shifting configurations of competing discourses where these positions are created, negotiated, resisted and even abolished. At this point, positionings that actors take are not constant but are continuously being re-shaped with how people position themselves with the available resources (material or conceptual tools) they bring to dynamics conversations, and in turn, the people are positioned with the emergence and dominance of values (often shaped by societal debates and visions that a particular society adapts to engage in the continual game of survival in relation to global and civilizational discourses). That is, the action of creating human agency under recent politics of competency discourse is, in fact, an exhibition of how dynamical educational actions make certain positions and spaces of actions possible through the invocation of a variety of moral orders, and of how society creates ways to allow actors to develop a storyline that appears reasonable and plausible to take. At the same time, the changing fields of moral orders around education create an arena for contestations for new positions to emerge. These new positions then try to seek their legitimacy by engaging in changing discursive acts that educational acts make available for actors to employ as a tool for justification. This does not mean that dominant positions will continue to stay dominant over time. These might shift and even be abolished and allow for new positions to emerge. Therefore, there are constant constellations of shifting positions under the nexus of discursive acts, storylines and the power dynamic enabled by the variety of moral orders.

## 4. Some Applications of Positioning Theory in Education

In the late 1800s, one dominant area in psychology focused on psychology as a laboratory-based experimental science (Green et al., 2020) [25]. In 2012, Harré [3] considered that positioning theory was a research program as part of cultural psychology, as opposed to purely experimental; his argument was that the importance of social representations of local moral orders are part of discursive practices. The location of positioning theory as a 'social constructionist approach' (Slocum & Van Langenhove, 2003) [32] pointed out the importance of language in the constitution of the social realm. In this way, positioning theory shares some connections to Vygotsky's work with important theoretical underpinnings (McVee et al., 2019) [33]. The inspiration in Vygotsky's work lies in the fact that speech acts and other actions are meditational tools applied to social situations. Here, positioning can be understood as a meditational tool as part of social activity (McVee et al., 2019) [33]. Moreover, Herbel-Eisenmann and Wagner (2010) [34], for example, have explored how power and authority relations are constituted within an education setting such as a mathematics classroom. Here, they revealed how authority structures are implicit and revealed through different positions that participants take in mathematics classroom practice, where students are assumed to have choices and obligations within complex discursive contexts taking place within dynamic discourse in the mathematics classroom.

Below, we have provided some key examples in relation to the application of positioning theory as an analytical lens in education and specifically in education for sustainable development (see Table 2). We have purposefully chosen these examples to illustrate how positioning theory is being interpreted within educational contexts.

**Table 2.** Examples of applications of positioning theory in education.

| Positioning Theory Application in Different Fields in Education | By Whom | Focus/Finding |
|---|---|---|
| Gendered identities | Anderson (2002, 2009) [22,35] | Investigation about gender identities in elementary classrooms to find out how gender, identity and literacy interactions occur. |
| Communication studies | Hirvonen (2016) [19] | Through an analysis of small-group dynamics from management board meetings in Finland, Hirvonen (2016) presents interactions of local moral orders in small groups. |
| Interracial | McVee (2011) [36] | Educational affordances of positioning theory for educational researchers and particularly for those working with literacy education and teacher education. |
| Mathematics education | Herbel-Eisenmann and Wagner (2010) [34] | Tracing the pronouns with other words that follow them, as a "lexical bundle". Interpreting the lexical bundles made them realize how the authority structures are implicit in mathematics classroom practice by understanding the ways in which students in mathematics are assumed to have choices in the discourse and obligations. |
| | Felix & Ali (2022) [23] | Tracing pronouns helped to understand a turning point in teachers collaborative work in mathematics. |
| English as a Second Language—education | Kayi-Aydar (2018, 2019) [17,37] | Presents how language in teachers from different backgrounds are being constructed professionally. Professional identities are being shaped through the ways teachers position themselves in relation to other teachers. |
| Science education | Ritchie (2002) [38] | While exploring the intersection of not only gender, but also status, this study explores how power is exercised in groups of students (grade 6). The conclusion is that students' previous experiences in social settings and the associated storylines enacted affect their present and future relationships. |
| Education for Sustainable Development | Felix et al. (2022) [39] Felix (2023) [40] | Identification of the use of different *we* pronouns in teachers' discourse, such as *humanitarian*, *institutional* and *classroom* [39]. Mostly, the uses of the pronoun *we* are situated in the context of duty within nature [39,40]. |
| Political identity studies | Slocum-Bradley (2009) [41] | Suggests an adaptation to the positioning theory triangle into a positioning theory diamond. This includes rights vs. duties, identities, storylines and also social forces. |

*Extending on One Example: Positioning Theory Applied to Environmental Discourses and Education for Sustainable Development*

One of the intriguing aspects is how "environmental" and "sustainability" moral orders are being integrated in the different levels characterized through Van Langenhove (2017) [27]. Initially studying environmental discourses, Harré et al. (1999) [42], examined the relationship between language and environmental thought, drawing principally upon linguistic theory. The linguistic approach was used to identify the means of persuasion in the discourse, i.e., different genres used in environmental discourses. The book includes an analysis of concepts in the language manifestations on studies conducted between 1992 and 1996. In the linguistic approach, they found out that pronouns can help to make sense of people's positions, particularly visible through the joint creation of a social act. This means that, through pronouns, it is possible to make sense how people are positioning themselves and how they are being positioned by others [1] (Harré & Van Langenhove, 1999; Van Langenhove, 2021) [1,43]. When tracing the pronoun "we" in environmental discourses, it can be specified as an inclusive (speaker + hearer) way. Another position that can occur in the "environmentalist discourse" is when someone uses a third voice position and legitimates a certain statement. An inference is that, through a third voice position, the implication of the individual as part of the environmental commitment is diluted.

For example, Felix (2023) [40], through interviews with primary teachers in Norway in relation to their engagement with education for sustainable development through critical thinking while using positioning theory, has found out the significance of considering critical thinking dispositions when teachers talk about nature and climate change. Here, she has found out how Norwegian primary teachers use *they* positions when it comes to identifying contexts such as Asia for doing more for the mitigating effects of climate change, while emphasizing as *we* in Norway are already acting towards mitigating climate change. Felix (2023) [40] has identified the importance of taking culture seriously to identify the specificity of moral orders constituted through egalitarian principles promoted within social democratic polity as Norway.

## 5. Summary

Positioning theory emerged after the experimental psychology period, especially in the 1980s. The main point has been following people's discourse that is framed in a positioning theory triangle, through speech acts, position and storylines. In this entry, we present the main concept in positioning theory through a brief historical overview. The main purpose of the entry is to offer an overview of the different approaches, shortly described and used in different subjects in education, especially in mathematics and in education for sustainable development. Positioning theory can offer a potential analytical tool to understand how different actors position themselves through employment of discursive acts within a social setting as education. Positioning theory will potentially enable us to gain deeper insights into how human actions are embedded within particular social settings, how they are constituted through performative educational acts entangled into different moral orders that define action and create conditions for educational action to take place, and how actors react to these performative actions.

**Author Contributions:** All authors contributed to the conceptualization, investigation, and analysis undertaken through this study. All authors have read and agreed to the published version of the manuscript.

**Funding:** This research received no external funding.

**Institutional Review Board Statement:** Not applicable.

**Informed Consent Statement:** Not applicable.

**Data Availability Statement:** Not applicable.

**Acknowledgments:** We would like to acknowledge the reviewers that helped us to improve the original manuscript.

**Conflicts of Interest:** The authors declare no conflict of interest.

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
