# Peer review of "Positioning Theory in Education"

_encyclopedia, doi:10.3390/encyclopedia3030073_

Round 1

Reviewer 1 Report

The text is very good. Thanks for the opportunity to read. 

  • avoid citation in the definition
  • maybe to add some information about how Van Langenhove & Harré, 1999 arrived to their triangle (some background interesting?)
  • if there are difference between discursive or narrative process, (55)
  • you talk of the teacher here without any introduction ! (57)
  • what is the link between Austen’s How to do things with words 66 (1962) and John Searle (1969) Speech Acts: An Essay in the Philosophy of Language and the triangle of Van Langenhove & Harré, 1999 ?
  • which context? sociology? psychology? (87)
  • social psychology modeled on experimental positivist (96) why only psychology? also sociology?
  • please connot  more rights and duties (103): moral? social? economical? gender? all? 
  • 107: ok moral of what morality? oriental?occidental?
  • 110: please introduce better the "teacher"example
  • 121: what is deontic powers?
  • 139: theory of moral order
  • 147: adaptation of variety of moral orders (Van Langenhove, 2017) adaptation to what? to education? by the authors? applyed in educational example? 
  • 152: social legitimacy? 
  • 190-194 very difficult and long
  • 226: please provvide how you reach the selections of the paper in "Table 2. Examples of applications of positioning theory in education": are paper from where?WHICH JOURNAL? Norway? Global South? any database? Google Schoolar?
  • 264: different subjects in education. -maybe just some (mathematic and SD)
  • 267: why only link with "human agency" is in fact formed through performative and not motivation or other concept? 

Author Response

Dear reviewer,

we wish to thank you for all your suggestions and comments. We have now uploaded the revised manuscript and addressing all the comments point by point. With king regards, the authors

Reviewer 2 Report

The issue appears very interesting and correlates with the narrative method of analysis. Positioning theory in practice can be a challenge for the educators, regardless of individual approach. However, it could be more broaden with the literature references and rooted deeper in philosophical area, dealing with e. g. Kolberg's theory, Habermass' s well.  The Author could analyze deeper the factors influencing the moral approach to life or education, what is also connected with the paradigm that should be more explicated and shown from different perspectives. What is more, it could be interesting to show the stages of moral acts of doing including correlation between affection (emotions), reason and the will fostering the deployment of this theory in a practical field that needs applying different educational methods and techniques. Thus, the sociological backgroung could be more developed in relation to the human integral, holistic development.  

Minor language improvement

Author Response

(The authors gave the same response as above.)

Reviewer 3 Report

The text is brilliant and toughtful. It definitely helps understand the positioning theory. However, a mis-contextualization of the positioning theory within the general frame of the social sciences has been found.

In order to improve the text, the authors are kindly invited to add 2/3 paragraphs at the beginning of the text by contextualizing the positioning theory in a specific moment (the 80s), and with references to:

- The relevant studies that brought the communication theory from a behaviouristic approach into a hemreneutic one (Bateson, Watzlawick)

- The critical approach to communication, and it relationship with power (Habermas)

- The performative component of language (Butler)

- The critical discourse analysis (Potter, Searle)

Author Response

(The authors gave the same response as above.)

Round 2

Reviewer 1 Report

ok

Reviewer 2 Report

After getting acquinted with the corrected text, particularly concerning the morality, and references, I feel satisfied with it, although at least one methodological reference could be added, but it is up to the Author. As I Have already mentioned I find the contents very interesting.